# An economic evaluation of the LINKEDin study: An intervention to reduce initial loss to follow-up among tuberculosis patients in South Africa

Michael Strauss[1]*, Muhammad Osman[2,3], Sue-Ann Meehan[2], Florian M. Marx[2,4,5], Anneke C. Hesseling[2], Andrew Boulle[6,7], Pren Naidoo[8], Gavin George[1,9]

1 Health Economics and HIV and AIDS Research Division, University of KwaZulu-Natal, Durban, South Africa, 2 Desmond Tutu TB Centre, Department of Paediatrics and Child Health, Faculty of Medicine and Health Sciences, Stellenbosch University, Cape Town, South Africa, 3 Faculty of Education, Health and Human Sciences, School of Human Sciences, University of Greenwich, London, United Kingdom, 4 Department of Infectious Disease and Tropical Medicine, Heidelberg University Hospital, Heidelberg, Germany, 5 DSI-NRF South African Centre of Excellence in Epidemiological Modelling and Analysis (SACEMA), Stellenbosch University, Stellenbosch, South Africa, 6 Centre for Infectious Disease Epidemiology and Research, Faculty of Health Sciences, School of Public Health and Family Medicine, University of Cape Town, Cape Town, South Africa, 7 Department of Health and Wellness, Health Intelligence Directorate, Western Cape Government, Cape Town, South Africa, 8 Public health Management Consultant, Cape Town, South Africa, 9 Division of Social Medicine and Global Health, Lund University, Lund, Sweden

* straussm@ukzn.ac.za

## Abstract

Tuberculosis (TB) patients who are diagnosed but not registered and initiated on TB treatment are categorised as initial loss to follow-up (ILTFU). ILTFU is a key driver of morbidity and mortality associated with TB and is a contributing factor to high TB transmission rates. LINKEDin was a quasi-experimental study which evaluated two specific interventions for reducing ILTFU in three high-TB burden provinces in South Africa, conducted from October 2018 to December 2020. As part of LINKEDin, we undertook a micro-costing from the healthcare provider perspective using an activity-based costing approach. Cost estimates included the cost of the operation of an integrated provincial health data centre in the Western Cape, apportioned to the TB activities it supported in the province. Cost estimates were linked to intervention outcomes to understand the incremental cost of the intervention per additional patient linked to care compared to rates of ILTFU in the absence of the interventions. Sensitivity analyses were conducted to account for uncertainty in the intervention outcomes, and for periods where the implementation of the intervention was interrupted due to COVID-19 related disruptions. Costing data were collected between August 2020 and March 2021. The total cost of implementing the LINKEDin intervention in the WC and KZN was $7 534.42 per month. The cost of implementing LINKEDin in the Western Cape accounted for 56% the total cost of the intervention – 8% from the operations of the PHDC and 48% from the cost of running the intervention – while

**Data availability statement:** The data are held in a public repository at Stellenbosch University, and are available at https://doi.org/10.25413/sun.30155884.

**Funding:** This research and publication were supported by the Bill and Melinda Gates Foundation (BMGF), INV- 007130. The contents are the responsibility of the authors and do not necessarily reflect the views of the BMGF. The funders had no role in study design, data collection and analysis, decision to publish, or preparation of the manuscript. ACH is financially supported by the South African National Research Foundation (NRF) through a South African Research Chairs Initiative (SARChI). The financial assistance of the NRF towards this research is hereby acknowledged. Opinions expressed, and conclusions arrived at, are those of the authors and are not necessarily to be attributed to the NRF.

**Competing interests:** The authors have declared that no competing interests exist.

only 44% of the total cost was accounted for by the intervention run in KwaZulu-Natal. The primary cost driver of the interventions were staff salaries, with the cost of data extraction and in-hospital activities low relative to primary healthcare (PHC)-based follow-up activities. In terms of cost effectiveness, the LINKEDin interventions in KZN was cost $377.28 per additional person linked to care, and $243.62 in the WC, per additional person linked to care. In the Western Cape, systematically tracking persons with TB using an automated system proved highly cost efficient compared to the more labour intense approach adopted in KwaZulu-Natal. Optimising the curation and management of data and increasing the effectiveness of tracing systems and processes can result in cost-savings.

## Introduction

South Africa has a high burden of tuberculosis (TB), with an estimated 270 000 people developing TB in 2023, with only 78% (211 810) registered and initiated on TB treatment [1]. Individuals who are lost from care between TB diagnosis and treatment initiation and registration are categorised as initial loss to follow-up (ILTFU) [2]. Previous South African studies have estimated ILTFU to be between 12% and 25% [3–5] with an estimated mortality of 17% among ILTFU patients [6]. Addressing the gap between TB diagnosis and treatment initiation and registration is vital to prevent ongoing transmission of TB [7] and decrease morbidity and mortality [8].

In South Africa, TB investigation, diagnosis, and treatment initiation take place at any level of care in the public healthcare system, but TB reporting systems are maintained at designated TB treatment sites. This includes primary healthcare (PHC) facilities, where persons with TB receive treatment on an outpatient basis, and specialised TB hospitals, where persons who require hospitalisation for TB are treated. In accordance with South African TB protocols, once a patient is tested for TB, the results are captured in the National Health Laboratory Service (NHLS) data system and recorded in an electronic TB register upon the initiation of TB treatment [2]. This typically occurs at a PHC facility. Patients not initiated, or who have started treatment but are not captured in the electronic TB register cannot be tracked. This can occur when a diagnosis is made in a hospital, and patients are referred to a PHC facility to initiate or continue treatment, or because patients who received a test at a PHC facility never return to the facility to receive their results.

The LINKEDin study was a quasi-experimental study conducted in three South African provinces—KwaZulu-Natal (KZN), the Western Cape (WC) and Gauteng Province (GP)—to investigate whether two health system strengthening interventions (hospital recording and an alert-and-response patient management intervention) could reduce ILTFU [9]. In South Africa, TB reporting systems are maintained at designated TB treatment sites, including specialized TB hospitals and all PHC facilities. In KZN and GP, LINKEDin utilised public sector TB reporting systems for implementation, however, in the WC, the intervention utilised data from the provincial health data centre (PHDC), which is housed in the provincial Department of Health and Wellness,

and harmonizes all electronic patient health data from all public sector services in the province, including disease specific reports for TB, laboratory results of TB tests, pharmacy and clinical records, TB treatment registers and other TB specific data recorded at hospital or PHC level [10].

The purpose of this economic evaluation was to estimate the total and comparative cost of the LINKEDin interventions for reducing ILTFU in KZN and the WC. We aimed to estimate the incremental cost of LINKEDin per additional TB patient linked to care (i.e., the cost per ILTFU averted) compared to status quo.

## Methods

### Intervention

The LINKEDin interventions aimed to reduce ILTFU among persons with TB (PWTB). ILTFU was defined as all persons with a confirmed Xpert MTB/RIF positive TB result for whom there was no evidence of linkage to a TB treatment facility for TB registration and treatment ≤30 days from date of diagnosis. The hospital-recording intervention aimed to improve recording of PWTB in hospitals; In KZN, newly diagnosed PWTB were identified from NHLS line lists of people with a positive TB test result. In the WC, the PHDC was used. Hospital staff confirmed whether patients were initiated on treatment in hospital. There were no additional interventions to assist patients to link to a TB treatment facility at discharge.

The alert-and-response patient management intervention, implemented at the PHC level, aimed to assist health personnel adopt a more systematic, patient level data system for monitoring and patient follow-up for those diagnosed in hospital and at PHC level. Specifically, the interventions set out to 1) implement an electronic TB register of all PWTB diagnosed in hospital (laboratory-confirmed and clinically diagnosed in the WC and laboratory-confirmed in KZN) thereby including non-reporting hospitals into the existing electronic TB notification system; and 2) implementing an alert-and-response patient management system to facilitate rapid linkage of PWTB to TB services at PHC facilities. In the WC, data in the PHDC enabled us to include clinically diagnosed patients, as the PHDC enabled us to track those not linked to care. In KZN, we could only identify people diagnosed with TB through the NHLS system (bacteriologically diagnosed). We could only identify clinically diagnosed PWTB from the TB treatment register (Tier.Net) which resulted in the exclusion of clinically diagnosed PWTB in KZN. In the WC, the PHDC was used to check for evidence of linkage to and registration at a TB treatment facility. There is an indicator in the PHDC which confirms TB registration. In KZN, matching algorithms were used to compare individuals with a TB diagnosis against Tier.Net. The data clerks would check the facility Tier.Net register for each PWTB. If they could not find them, the co-ordinator would check the district Tier.Net file, in case they accessed care at another facility. All PWTB eligible to link but with no evidence of linkage were followed up by a short message service (SMS), followed by a phone call and then creating a referral for a community-based health worker (CHW) to do a home visit to facilitate linkage. PWTB who had no telephonic details were immediately referred to a CHW. [9].

Fig 1 Shows the flow of patients in the LINKEDin intervention and provides a framework for estimating the costs of the intervention activities.

### Costing perspective, approach, and data collection

An estimation of the costs of implementing the interventions was undertaken from a health-care provider perspective. Additionally, we estimated the costs of operating a provincial health data centre, based on the operation of the PHDC in the WC in so far as it contributed to the TB programme, and its utility to the implementation of the intervention in the WC.

A mixed-methods approach was used to identify, measure and value costs [11]. Costs were identified and valued using a bottom-up micro-costing approach where data were extracted from expenditure reports, accounting records and equipment inventories of implementing partners, with the remainder of the costs estimated using a top-down gross-costing approach. Cost and resource use data for the PHDC focused on the operational cost of the intervention activities,

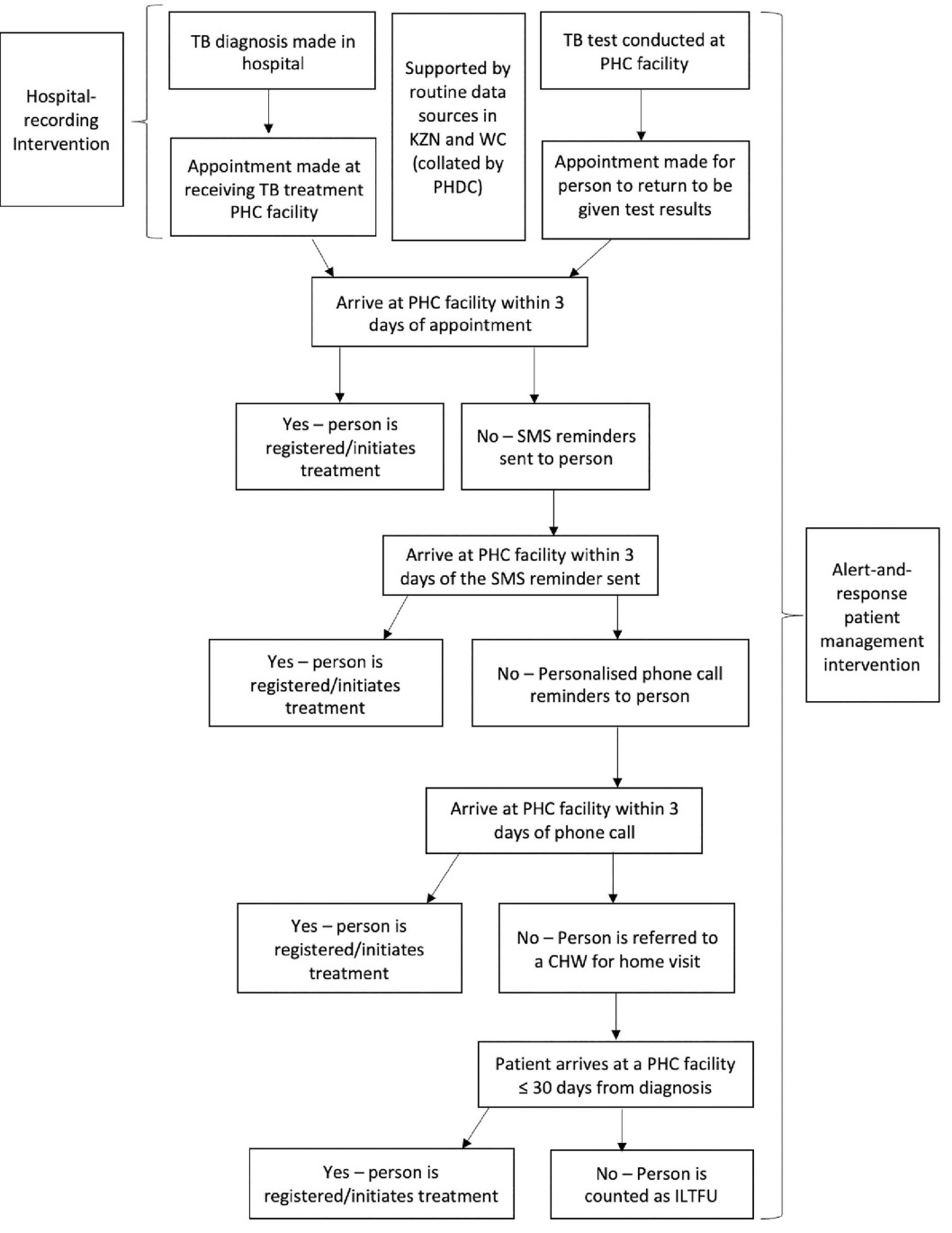

TB=Tuberculosis, KZN=KwaZulu-Natal province, WC=Western Cape province, PHDC, Provincial Health Data Centre, PHC=primary healthcare, SMS=short message service, CHW=community healthcare worker, ILTFU=initial loss to follow up

**Fig 1. LINKEDin intervention patient flow.**

excluding start-up and research costs. Resources were allocated in line with an activity-based approach [12,13] according to their economic classification, as per the activities identified. Costing data were collected from August 2020 to March 2021. At the end of the intervention, data were extracted from the electronic health records systems in KZN and the WC covering the 3-month baseline period (October 2018 to December 2018) and the intervention period (January 2019 to December 2020).

**The cost of implementing the LINKEDin intervention in the Western Cape and KwaZulu-Natal.** The cost of implementing the interventions depended on several key economic inputs, including staff, consumables, equipment, building/office space, training, and facility management and supervision. Resource use was estimated based on the key intervention activities that were conducted at each of the study sites in the WC and KZN. Table 1 shows the list of intervention activities for the primary component of the interventions in each of the provinces.

At the facility level, staff members, including data clerks and HCWs tasked with processing the data used for tracking and managing patients, were trained on the new systems – including how to access the PHDC system for those staff in the WC. These staff costs were calculated at the facility level (hospitals and PHC facilities), including training, salaries, equipment, and other costs relating to staffing capacity. Staff costs were estimated using actual staff salaries where possible, and public sector salary scales where these costs were not directly available. For activities directly related to patient follow-up, costs were estimated based on the amount of time staff allocated per patient. For activities not directly related to patient follow-up (e.g., checking patients against hospital data records, identifying PWTB who are eligible to link to a PHC facility, drawing up and comparing lists, sending lists to the relevant people), staff costs were estimated based on the level of effort (number of hours per month) that staff spent on each of the intervention activities. LINKEDin project supervisors in each province were asked to monitor staffing levels of effort per activity and provide average estimates of the number of hours staff spent on a monthly basis per activity. Staffing costs not

**Table 1. Intervention activities in the Western Cape and KwaZulu-Natal.**

| KWAZULU-NATAL | | | WESTERN CAPE | | |
|---|---|---|---|---|---|
| **Category** | **Activity** | **Description** | **Category** | **Activity** | **Description** |
| Data Extraction | Activity A: | Use the Xpert alerts to identify newly diagnosed persons with TB<br>-in hospital<br>-in PHC facilities | Data Extraction | Activity A: | Use the public health care system TB reporting systems to identify newly diagnosed persons with TB<br>-in hospital<br>-in PHC facilities |
| Hospital Recording Intervention | Activity B.1: | **GJ Crookes** Checking patients against hospital data records to identify patients not started on TB treatment.<br>**Gamalakhe** Checking patients against facility data records to identify patients not started on TB Rx. | Hospital Recording Intervention | Activity B.1 | Send list of persons with TB (alive and admitted) to IPC, highlight those not started on TB treatment. |
| | | | | Activity B.2: | **Khayelitsha** Visit persons with TB in hospital for 'enhanced engagement' prior to discharge |
| Alert and Response Intervention | Activity C: | Identify persons with TB who are eligible to link to a PHC facility, have not linked and require systematic follow-up/Using Tier | Alert and Response Intervention | Activity C: | Identify persons with TB who are eligible to link to a PHC facility, have not linked and require systematic follow-up |
| | Activity D: | Send SMS to identified persons with TB | | Activity D: | Send SMS to identified persons with TB |
| | Activity E: | For those who remain unlinked, make telephonic contact with persons with TB | | Activity E: | For those who remain unlinked, make telephonic contact with persons with TB |
| | Activity F: | For those who remain unlinked, refer to CHW for home visit | | Activity F: | For those who remain unlinked, refer to CHW for home visit |
| Training | Activity G: | Training of facility staff to do these interventions | Training | Activity G: | Training of facility staff to do these interventions |

related to intervention activities were excluded. Other resource costs were estimated using LINKEDin study budgets and expenditure reports. Staffing capacity to perform intervention tasks was not evaluated, rather a total resource input requirement per intervention activity and per TB patient was estimated to understand how the staffing resource requirement links to reductions in the number of patients ILTFU.

Equipment costs were calculated based on the year of purchase and annualized using a standard discount rate of 3% and the years of useful life data provided on the South African Revenue Service website [14]. Annual equipment costs were then converted to monthly costs and allocated across activities based on the level of effort allocated to each activity by the staff member using the equipment.

Training was conducted at inception as well as throughout the course of the intervention. Once-off training costs were annualized, conservatively assuming the inception training would need to be conducted once every five years. Continuous training costs were included in the operating costs of the intervention. The cost of consumables was apportioned to activities based on the level of staff effort across the intervention activities, and the consumables used by each individual staff member.

Building space in the WC and in KZN (including office space at implementing partners' main office and rooms in health facilities used by HCWs and data clerks) was rented, with rental costs including maintenance and utilities. Building space costs were apportioned to the various intervention activities using the level of effort of each staff member allocated across each activity.

Overall management and overhead costs associated with running the intervention related mostly to data and administrative support (including human resources and financial management) were estimated and apportioned between the WC and KZN based on the level of effort of staff and utilisation of other resources.

**Estimating the cost of operating a data centre.** Cost estimates of the WC PHDC were determined by identifying and valuing the resource requirements for their operations. Where possible, cost estimates related to each individual resource component were extracted from expenditure records, equipment inventories, routine reports, other accounting records, and salary information. Fixed costs of running the PHDC were annualized using a conventional economic costing approach and guided by experts at the PHDC as to the appropriate lifespan of resources such as infrastructure and equipment [12,14].

The WC PHDC curates and manages data for several critical health services, not only TB. Thus, the total cost of running the data centre was apportioned to TB activities based on the full range of disease categories the PHDC serves and in consultation with experts at the data centre. The costs of the PHDC for supporting the LINKEDin study were further apportioned to the LINKEDin specific facilities.

**Linking costs to outcomes.** We related overall costs for implementing the LINKEDin interventions to the estimated number of people diagnosed with TB in whom ILTFU was averted through LINKEDin (i.e., the additional number of people linked to TB treatment). Uncertainty in the estimates of ILTFU over the course of the intervention is captured by bootstrapping intervention outcomes, where 200 parameter sets were sampled from all uncertainty ranges. For the outcomes, we used mean, low and high estimates, with low and high values calculated as the 2.5th and 97.5th percentile of resultant outcomes [15] (S1 Table).

The interventions in the WC and in KZN resulted in a reduction of patients with ILTFU. In KZN, 1999 individuals were diagnosed with TB, of which 285 (14%) were ILTFU. In the WC, 9359 individuals were diagnosed with TB, of which 1664 (18%) were ILTFU [9]. In the first quarter of 2019, the intervention began rolling out in all sub-districts, but it took some time for all components of LINKEDin to be fully operationalised. Further to this, in the second quarter of 2020, South Africa initiated a series of strict lockdowns as part of a national response to the coronavirus 2019 (COVID-19) epidemic, which had a significant impact on patients' ability to access health facilities, resulting in lower numbers of people being diagnosed with TB (as well as lower absolute numbers of ILTFU) recorded in the months within the lockdown period [16]. Further details on the roll-out of the interventions can be found elsewhere [9].

We calculated incremental cost-effectiveness ratios (ICERs) as the incremental costs for implementing (and operating) the LINKEDin interventions divided by the number of additional patients linked to TB care (i.e., the reduction in ILTFU) compared to the absence of the intervention (measured over the baseline period before LINKEDin was implemented). Costs during the baseline period for calculating ICERs were assumed to be zero, given that the intervention had not started. The number of people linked to care during the baseline period was used as a comparator for the calculation of the ICERs.

$$ICER = \frac{additional\ cost\ of\ the\ intervention}{additional\ number\ of\ people\ linked\ to\ care\ because\ of\ the\ intervention}$$

The cost estimates of the interventions are per month in 2019, establishing the average number of additional PWTB linked to care per month by sub-district in KZN and the WC. Cost data were collected in South African rands (ZAR), and then converted to US Dollars, using the average exchange rate over the course of the implementation of the intervention (ZAR15.62 = US$1) in 2019 [17].

**Ethics statement.** This costing study was based on healthcare system cost estimates and did not include personal identifiable data. The study used aggregate estimates of ILTFU obtained from the LINKEDin study. The LINKEDin study was approved by the Health Research Ethics Committee at Stellenbosch University (N18/07/069), the University of the Witwatersrand (M190128), and the relevant provincial departments of health. It was also approved by the KwaZulu-Natal Department of Health (NHRD ref: KZ_201808_053), and the Western Cape Department of Health (NHRD ref: WC_201808_034) and was conducted according to the guiding principles within the Declaration of Helsinki.

## Results

### Monthly intervention costs

The total cost of implementing the LINKEDin intervention in the WC and KZN was $7 534.42 per month. The cost of implementing LINKEDin in the Western Cape accounted for 56% the total cost of the intervention—8% from the operations of the PHDC and 48% from the cost of running the intervention—while only 44% of the total cost was accounted for by the intervention run in KwaZulu-Natal. Staffing costs were a key driver of the costs associated with running the intervention, accounting for 87% of the total cost. Equipment, consumables, and buildings accounted for 5%, 6% and 2% of the total costs respectively (Table 2).

A large portion of the cost of running the interventions in both the WC and KZN came from the management and overhead costs linked to the interventions. This was partly because oversight efforts were relatively intensive in the LINKEDin study to ensure that the roll-out of the interventions were responsive to context specific factors that would determine the success of the interventions in each province. In the WC, overhead and management costs accounted for 32% of the total monthly running cost of the interventions. In KZN, this was lower at 20% (Table 3).

Table 2. Total monthly intervention costs by province and cost category.

| Cost Category | KwaZulu-Natal | | Western Cape | | | | LINKEDin Total | |
| --- | --- | --- | --- | --- | --- | --- | --- | --- |
| | LINKEDin | % | PHDC | % | LINKEDin | % | KZN & WC | % |
| Monthly Staff Cost | $ 2 950.23 | 89% | $ 527.90 | 85% | $3 099.60 | 86% | **$6 577.78** | **87%** |
| Monthly Equipment Cost | $ 61.33 | 2% | $ 7.95 | 1% | $ 271.09 | 8% | **$ 340.38** | **5%** |
| Monthly Consumables Cost | $ 228.22 | 7% | $ 58.84 | 10% | $ 153.79 | 4% | **$ 440.85** | **6%** |
| Monthly Buildings Cost | $ 64.06 | 2% | $ 23.98 | 4% | $ 87.37 | 2% | **$ 175.41** | **2%** |
| TOTAL | **$ 3 303.83** | | **$ 618.67** | | **$3 611.85** | | **$7 534.42** | |
| % of LINKEDin Total Cost | **44%** | | **8%** | | **48%** | | | |

**Table 3. Cost per month by intervention activity and province.**

| | KwaZulu-Natal | | Western Cape | | GRAND TOTAL | |
|---|---|---|---|---|---|---|
| **LINKEDin** | | | | | | |
| **DATA EXTRACTION** | | | | | | |
| Activity A (Data extraction) | $ 418.52 | (13%) | $ 297.92 | (7%) | $ 716.44 | (10%) |
| **HOSPITAL RECORDING INTERVENTION** | | | | | | |
| Activity B (Identification of ILTFU) | $ 809.96 | (25%) | $ 629.76 | (15%) | $ 1,439.72 | (19%) |
| **ALERT AND RESPONSE INTERVENTION** | | | | | | |
| Activity C (Identification of ILTFU) | $ 596.55 | (18%) | $ 144.05 | (3%) | $ 740.60 | (10%) |
| Activity D (Send SMS) | $ 156.91 | (5%) | $ 308.00 | (7%) | $ 464.91 | (6%) |
| Activity E (Phone call) | $ 156.91 | (5%) | $ 409.88 | (10%) | $ 566.79 | (8%) |
| Activity F (CHW home visit) | $ 180.44 | (5%) | $ 308.00 | (7%) | $ 488.45 | (6%) |
| **TRAINING** | | | | | | |
| Activity G (Training) | $ 336.53 | (10%) | $ 172.45 | (4%) | $ 508.98 | (7%) |
| **CROSS-CUTTING OVERHEAD AND MANAGEMENT COSTS** | $ 648.02 | (20%) | $ 1,341.79 | (32%) | $ 1,989.81 | (26%) |
| **PHDC OPERATING COSTS** | | | $ 618.67 | (15%) | $ 618.67 | (8%) |
| **TOTAL** | **$ 3 303.83** | | **$ 4 230.53** | | **$ 7 534.36** | |

The cost of data extraction activities accounted for 10% of the monthly cost of the interventions, while the monthly cost of the hospital recording intervention, and the alert and response intervention accounted for 19% and 30% respectively. Training costs constituted a much lower proportion, just 7% of the total cost per month. Total cross-cutting overhead and management costs accounted for a large proportion of the interventions. These costs would likely reduce significantly if the interventions were institutionalised and run within the existing structures of the provincial departments of health, supported by the PHDC.

## Monthly cost of operating the PHDC in the Western Cape

The total monthly operating cost of the PHDC including overhead costs was estimated to be $78 740.31, with staffing costs the primary cost driver, contributing 85% of the total cost (S2 Table). The PHDC activities relating to TB accounted for approximately 22% of the PHDC's operations, with these activities supporting 420 facilities across the province including the facilities in this study. This means that the cost of the PHDC per TB facility, supported with TB data processing, is approximately $41.25 per month. Given that the PHDC supported a total of 15 facilities in the WC for the delivery of the LINKEDin interventions, the total cost of PHDC support apportioned to LINKEDin was $618.67 per month, approximately 8% of the total cost of the intervention in the Western Cape.

## Incremental Cost Effectiveness Estimates

The cost of the LINKEDin interventions in KZN was $377.28 (low estimate $302.82; high estimate $514.84) per additional person linked to care, and $243.62 (low estimate $188.75; high estimate $321.21) in the WC, per additional person linked to care (see Table 4).

By sub-district, we found that the intervention in Tygerberg had the lowest cost per additional person linked to care, but with the lowest reduction in ILTFU. Only the hospital recording intervention was implemented in Tygerberg, and although this facility saw the smallest reduction of patients ILTFU, implementation costs were lower, resulting in a lower cost per additional person linked to care. In all sub-districts except Tygerberg, the interventions had a lower ICER in the months when it was being fully implemented (i.e., excluding the start-up period and COVID-19 shutdown period).

**Table 4. Incremental cost effectiveness of the interventions in KZN and the WC by province and sub-district, including average, high and low estimates of the number of additional patients linked to care per month.**

| Intervention Period (Jan 2019-Dec 2020) | | Total | Ray Nkonyeni SD | Umdoni SD |
|---|---|---|---|---|
| KZN | Additional persons with TB linked to care/month | 8.75 | 6.09 | 2.67 |
| | Additional persons with TB linked to care/month (low and high estimates) | (6.42–10.91) | (4.81–7.2) | (1.6–3.71) |
| | Additional cost/month | $ 3,303.83 | $ 1,998.16 | $ 1,305.67 |
| | **ICER** | **$ 377.42** | **$ 328.28** | **$ 489.57** |
| | ICER (low and high estimates) | ($302.82 – $514.84) | ($277.59 – $415.16) | ($351.75 – $813.89) |
| Excluding Start-up Period and COVID-19 Shutdown | | | | |
| KZN | Additional patients linked to care/month | 10.27 | 6.73 | 3.54 |
| | Additional patients linked to care/month (low and high estimates) | (7.76–12.64) | (5.37–8.06) | (2.38–4.58) |
| | Additional cost/month | $ 3,303.83 | $ 2,002.05 | $ 1,301.78 |
| | **ICER** | **$ 321.76** | **$ 297.38** | **$ 368.18** |
| | ICER (low and high estimates) | ($261.31 – $425.82) | ($248.25 – $372.49) | ($284.3 – $546.09) |
| Intervention Period (Jan 2019-Dec 2020) | | Total | Khayelitsha SD | Tygerberg SD |
| WC | Additional patients linked to care/month | 17.37 | 10.47 | 6.90 |
| | Additional patients linked to care/month (low and high estimates) | (13.17–22.41) | (9.91–13.65) | (3.26–8.76) |
| | Additional cost/month | $ 4,230.59 | $ 3,514.91 | $ 715.68 |
| | **ICER** | **$ 243.62** | **$ 335.85** | **$ 103.72** |
| | ICER (low and high estimates) | ($188.75 – $321.21) | ($257.41 – $354.78) | ($81.71 – $219.31) |
| Excluding Start-up Period and COVID-19 Shutdown | | | | |
| WC | Additional patients linked to care/month | 18.65 | 11.20 | 7.44 |
| | Additional patients linked to care/month (low and high estimates) | (14.48–25.55) | (10.55–15.53) | (3.93–10.02) |
| | Additional cost/month | $ 4,230.59 | $ 3,514.91 | $ 715.68 |
| | **ICER** | **$ 226.89** | **$ 313.71** | **$ 96.18** |
| | ICER (low and high estimates) | ($165.6 – $292.14) | ($226.31 – $333.05) | ($71.46 – $182.21) |

## Discussion

This study analysed the costs of implementing interventions aimed at reducing ILTFU among persons newly diagnosed with TB in South Africa. In this study, the cost per additional person linked to care was $337.28 in KZN and $243.62 in the WC. The interventions proved less costly per patient linked to care in the WC than in KZN due in part to the more automated process and higher number of PWTB, and subsequent linkages. While some of the costs were directly linked to the overall number of patients (for example the cost of SMSes, and in person follow-ups by CHWs), some intervention activities exhibited economies of scale, because the same resources were required regardless of the number of people ILTFU.

The primary cost driver of the interventions, including the operation of the PHDC, was attributed to staff salaries. In the WC, staffing costs were lower as a percentage of the total costs, which was partly due to some of the processes being partially or fully automated. Fully automated track and trace systems that require little input from hospital and clinic staff could significantly decrease the cost of reducing ILTFU. This approach would also be consistent with the South African national digital health strategy [18].

The study further revealed that the cost of data extraction and in-hospital activities was small in comparison to community-based follow-up activities. In the WC, automation of LINKEDin activities using the PHDC supported both in the hospital-based intervention—identifying ILTFU patients—and in the community-based follow-up—identifying ILTFU patients, sending automated text messages and automated follow-up phone calls. This use of an automated system proved highly cost efficient compared to the more labour-intensive approach adopted in KZN, which relied on intervention staff as well as hospital and PHC staff to conduct these activities. This highlights the importance of data systems that

are increasingly automated and updated in real-time, and which have the potential to reduce the burden on facility staff of identifying and following patients who are ILTFU. While this of course requires significant set-up costs, the operational cost offers cost savings and lower rates of ILTFU over time.

While LINKEDin was found to be effective in reducing ILTFU, some patients remained unreachable during the tracing and outreach process which was resource intensive. Although previous research has shown that active case finding interventions for PWTB can in themselves be cost effective in spite of the fact that not every patient can be reached through this kind of intervention [19], one of the key questions this raises is the relative importance of the utilisation of CHW home visits compared to improvements in the automation of data and systems in facilities and in centralised data repositories. Improving the quality of real-time data has the potential to reduce ILTFU reports and to improve the efficiency of targeted follow-up with patients who need to be linked to care by facility staff and CHWs.

To our knowledge, this study is the first to provide an economic evaluation of interventions that specifically address ILTFU in South Africa and adds to a limited body of costing analysis undertaken on interventions aimed at improving outcomes along the TB care cascade. Further research and analysis on the impact of alternative interventions, combined with an economic evaluation, is essential in assessing whether the added costs are reasonable when compared to the ILTFU outcomes achieved in this study. Modelling studies have shown the potential cost effectiveness of interventions to increase linkage to care among PWTB as well as decreased onward transmission and reduction in morbidity and mortality [20].

While it is not possible to determine whether this intervention is cost-effective without consensus on a cost-effectiveness threshold in South Africa and without quality-adjusted life year (QALY) or disability-adjusted life years (DALY) estimates linked to reductions in ILTFU, investing in strengthening health systems to support the management of TB is likely to be a high value investment [21]. A central tenet of any intervention remains scalability and sustainability, which is beyond the scope of this study. A recent analysis of the burden of disease caused by incident TB including post-TB morbidity and mortality, estimated health losses of 8.46 DALYs per TB incident case among treated TB cases, compared to 22.09 DALYs per TB incident case among untreated TB cases [22].

Although some patients may eventually link again to care after being ILTFU without intervention, and others who are linked to care because of interventions such as LINKEDin but may be lost from care later on (or indeed not complete treatment), these results suggest that there are health outcomes which can be averted by reducing ILTFU among people living with TB. Given the low cost of LINKEDin per ILTFU averted, and the potentially large number of DALYs averted, it is highly likely that LINKEDin would be cost-effective using almost any acceptable threshold, although more data would be required to estimate specific values. While the LINKEDin intervention was effective in both KZN and WC sites, leveraging a consolidated health data environment in the WC led to increased efficiency, which could be further increased if the data environment was optimized to collect and hold more current data about patients, which would increase the efficiency of recall.

An important limitation of this study is that the reported outcomes are difficult to link to final health outcomes such as morbidity and mortality. Although some morbidity and mortality data were collected among patients who initiated treatment, it is unclear what the final outcomes might be among the patients that remained ILTFU. Some patients who remain untreated might die, while others may recover enough to delay treatment. Some patients could link to treatment at a different facility and potentially in a different sub-district or even province. The length of the delay until a patient eventually links to treatment or dies will also influence onward transmission and resulting health outcomes. This means that estimating aggregated health outcomes (such as DALYs or QALYs) based on data from the LINKEDin interventions would at best be difficult, and at worst produce misleading results for decision-making. In KZN, facilities function differently within and across districts, and across urban and large rural settings, so our findings from one district may not be generalizable across the entire province.

Although an integrated, data-supported interventions come at a low cost and are likely to be cost-effective, questions remain about how these interventions could be scaled up across South Africa. First, the interventions would need to be repackaged to maximise health outcomes and minimise costs in contexts from other provinces, particularly in provinces without a data centre. Second, as data systems are improved, there will be scope for increased automation of the intervention, which can reduce the high human resource cost of linking patients to care. In this study, reductions in transmission, morbidity and mortality were not explicitly linked to reductions in ILTFU and future research could investigate the implications in terms of the potential health benefits when the intervention is delivered at scale. Finally, further analysis is needed to understand the budget impact and how cost-effectiveness changes when the intervention is delivered at scale.

## Conclusion

The findings from this analysis suggest that the cost of reducing ILTFU is low, and that the benefits of support from a data centre (including access to better quality data and reduced lags for identification of patients ILTFU) come at a similarly low cost. Optimising the curation and management of data can result in cost-savings. The ability of electronic data systems to avail real-time information to facilities and the CHWs tasked with following up with patients could eliminate unnecessary wastage of human and financial resources. The PHDC proved instrumental in identifying which persons with TB had not linked to care and required follow-up, but importantly, these data need to be relayed to TB clerks or information officers at a health facility in a timely manner for the appropriate action to be taken. However, without effectively managed data systems, the optimal cost benefits will not be realised.

## Supporting information

**S1 Table. Incremental effectiveness of the interventions compared to baseline.**
(DOCX)

**S2 Table. Monthly operating cost of the PHDC in the Western Cape.**
(DOCX)

## Acknowledgments

We wish to acknowledge Interactive Research and Development (IRD) in KwaZulu Natal Province and the University of Cape Town and the Centre for Infectious Disease Epidemiology and Research (CIDER) in the Western Cape Province. We further acknowledge the staff at the Western Cape Provincial Health Data Centre (PHDC) for their invaluable assistance. A specific thank you to Jolene Chetty and Rosemary Foster who supported data collection.

## Author contributions

**Conceptualization:** Michael Strauss, Muhammad Osman, Sue-Ann Meehan, Gavin George.

**Data curation:** Michael Strauss, Muhammad Osman, Sue-Ann Meehan.

**Formal analysis:** Michael Strauss, Gavin George.

**Funding acquisition:** Sue-Ann Meehan.

**Investigation:** Michael Strauss.

**Methodology:** Michael Strauss, Gavin George.

**Project administration:** Sue-Ann Meehan.

**Supervision:** Sue-Ann Meehan.

**Validation:** Muhammad Osman, Sue-Ann Meehan, Florian M Marx, Anneke C Hesseling, Andrew Boulle, Pren Naidoo.

**Writing – original draft:** Michael Strauss, Muhammad Osman, Sue-Ann Meehan, Gavin George.

**Writing – review & editing:** Michael Strauss, Muhammad Osman, Sue-Ann Meehan, Florian M Marx, Anneke C Hesseling, Andrew Boulle, Pren Naidoo, Gavin George.

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
