## [Decision Letter · Decision Letter 0]

30 Jul 2025

PONE-D-24-58934An Economic Evaluation of the LINKEDin study: An intervention to reduce initial loss to follow-up among tuberculosis patients in South AfricaPLOS ONE

Dear Dr.  Strauss,

Thank you for submitting your manuscript to PLOS ONE. After careful consideration, we feel that it has merit but does not fully meet PLOS ONE’s publication criteria as it currently stands. Therefore, we invite you to submit a revised version of the manuscript that addresses the points raised during the review process.

We look forward to receiving your revised manuscript.

Kind regards,

Ebenezer Wiafe, PhD, MPharm, Pharm D

Academic Editor

PLOS ONE

Journal Requirements:

3. In the online submission form, you indicated that “Data from this study are available to researchers who meet the criteria for access to confidential data and is available from the study PI, Sue-Ann Meehan, available at sueannm@sun.ac.za. Co-investigators are ethically bound to safeguard study materials, including data. Data usage by anyone external to listed co-investigators of the study, requires permission from the Health Research Ethics Committee of Stellenbosch University.”

5. Please include your tables as part of your main manuscript and remove the individual files. Please note that supplementary tables (should remain/ be uploaded) as separate "supporting information" files.

Additional Editor Comments:

None at the moment.

Reviewers' comments:

Reviewer's Responses to Questions

**Comments to the Author**

1. Is the manuscript technically sound, and do the data support the conclusions?

Reviewer #1: Yes

Reviewer #2: Yes

2. Has the statistical analysis been performed appropriately and rigorously? 

Reviewer #1: Yes

Reviewer #2: Yes

3. Have the authors made all data underlying the findings in their manuscript fully available?

Reviewer #1: Yes

Reviewer #2: Yes

4. Is the manuscript presented in an intelligible fashion and written in standard English?

Reviewer #1: Yes

Reviewer #2: Yes

5. Review Comments to the Author

Reviewer #1: The paper is well written, with clear language and addressing a very important topic. The study has been conducted at two high-burden TB provinces with differences in the manner that TB services are managed. It is noted that clinically diagnosed patients were not included in the study in KZN province. This study was conducted during a difficult period, during Covid-19 pandemic. The authors have explained how they managed to continue the intervention during this period. The study included a few sites per province. Staffing constitutes 85 % of the total cost. The cost in WC was found to be lower than KZN. WC had more patients overall than KZN which partially explains the cost and also KZN had a lot of laborious operations unlike WC that had automated tasks.

General comments:

I encourage the authors to reflect critically on the different manner that TB services are provided in these 2 provinces. Staff cost is the cost driver. I would like to understand which staff are referred to. Are these TB clerks or general administration clerks. If these are TB appointed staff members, what was the staffing situation in WC during the study and now that probably the situation has changed. The TB programme has been fully integrated into other services. One can say there is no provincial or provincial TB programme in WC. If this is correct, are these results still applicable? How can this intervention be introduced in the context of decreasing number of staff?

Specific comments:

Abstract:

The abstract is good, it provides the summary of the work done, although it does not provide the main findings of the study and the respective costs per province.

Introduction:

Line 25: use the 2024 Global TB Report estimates which is 270,000 for year 2023. Please say something about the provision of TB services overall in these provinces. I am happy with the content covered ion the introduction.

Methods:

Lines 70 - 74 provide eligibility criteria for each province. Please explain why clinical diagnosed patients were not included in KZN. How is the TB notification done in the facilities used for this intervention? Please describe briefly how access to TIER.Net was in KZN.

Discussion:

- The main issue to be added here is the one on staff mentioned as per my general comments and the new working arrangements in the WC province. It is not clear whether these findings are still applicable given the restructuring in WC.

- Additionally, please clarify whether the findings may be generalized within KZN given the differences between facilities even in the same district.

Reviewer #2: This study presents a relevant and well-structured evaluation of interventions aimed at reducing initial loss to follow-up (ILTFU) among TB patients in South Africa. Below are some few comments to improve the manuscript.

Abstract

• State the period the study was carried out

• The two specific interventions evaluated are not described in sufficiently. Briefly outlining what each intervention entailed would provide clearer context for the cost and outcome comparisons.

• While cost drivers are mentioned, the abstract would benefit from including actual cost figures or incremental cost-effectiveness ratios to enhance reader’s understanding of the magnitude of cost differences.

Methods

• Page 11, Line 184: “…1999 individuals were diagnosed with TB were diagnosed…”. Please chek this sentence.

The cost of implementing the LINKEDin intervention in the Western Cape and KwaZulu-Natal

• Page 8, line 121-123: Given that the interest is on incremental or additional costs for implementing the intervention, why do you include the salary of staff(already born by the government) in your staff cost calculations? I thought the best will be to include only additional payments made to the staff, maybe as motivation, due to the intervention.

• Page 9, line 127-128: Please explain further how you collected the data on the level of effort (number of hours per month) that staff spent on each of the intervention activities. Was it collected daily, weekly, monthly etc?

• Page 12: Calculation of ICER – The explanation of the cost calculations lacks sufficient detail. Specifically, more information is needed on the baseline costs before the implementation of LINKEDin.

• It would have been good to conduct a Budget Impact Analysis to inform scale up decision.

Results

Monthly intervention costs

The figures presented in the tables do not match those in the text; therefore, it is necessary to cross-check and ensure consistency between them. For instance:

• Page 13, line 216: I cannot find $14 023.69 in Table 2.

• Page 13, line 221: I cannot find 88% in Table 2.

• Page 13, line 222: I cannot find 4%, 3% in Table 2.

• Page 13, line 227-241: Since cost category proportions relative to total costs are frequently mentioned, it would be helpful to include the percentage contribution of each cost category to the total costs in Table 3

6. PLOS authors have the option to publish the peer review history of their article (what does this mean?). If published, this will include your full peer review and any attached files.

Reviewer #1: No

Reviewer #2: **Yes:**Dr Maxwell Dalaba

---

## [Author Response · Author response to Decision Letter 1]

26 Sep 2025

Dear Reviewers,

Thank you for the thoughtful comments which have helped to strengthen our manuscript. We have responded to each comment in the document attached. The line numbers referred to correspond to the line numbers in the manuscript with track changes.

---

## [Decision Letter · Decision Letter 1]

27 Jan 2026

An Economic Evaluation of the LINKEDin study: An intervention to reduce initial loss to follow-up among tuberculosis patients in South Africa

PONE-D-24-58934R1

Dear Michael Strauss,

We’re pleased to inform you that your manuscript has been judged scientifically suitable for publication and will be formally accepted for publication once it meets all outstanding technical requirements.

Kind regards,

Ebenezer Wiafe, PhD, MPharm, Pharm D

Academic Editor

PLOS One

Additional Editor Comments (optional):

Reviewers' comments:

Reviewer's Responses to Questions

**Comments to the Author**

1. If the authors have adequately addressed your comments raised in a previous round of review and you feel that this manuscript is now acceptable for publication, you may indicate that here to bypass the “Comments to the Author” section, enter your conflict of interest statement in the “Confidential to Editor” section, and submit your "Accept" recommendation.

Reviewer #1: All comments have been addressed

Reviewer #2: All comments have been addressed

2. Is the manuscript technically sound, and do the data support the conclusions?

Reviewer #1: Yes

Reviewer #2: Yes

3. Has the statistical analysis been performed appropriately and rigorously? 

Reviewer #1: Yes

Reviewer #2: Yes

4. Have the authors made all data underlying the findings in their manuscript fully available?

Reviewer #1: Yes

Reviewer #2: Yes

5. Is the manuscript presented in an intelligible fashion and written in standard English?

Reviewer #1: Yes

Reviewer #2: Yes

6. Review Comments to the Author

Reviewer #1: All issues raised in previous review have been adequately addressed. A clean copy provided a comprehensive document with all tracked changes accepted.

The document with tracked changes reflects changes made to the manuscript. I went through point by point, and I am satisfied with the responses from the authors and the modifications made in the revised manuscript.

Reviewer #2: The authors have adequately addressed concerns raised. They should re through the manuscript and corect typos.

7. PLOS authors have the option to publish the peer review history of their article (what does this mean?). If published, this will include your full peer review and any attached files.

Reviewer #1: No

Reviewer #2: **Yes:**Maxwell Ayindenaba Dalaba

---

## [Editor Report · Acceptance letter]

PONE-D-24-58934R1

PLOS One

Dear Dr. Strauss,

I'm pleased to inform you that your manuscript has been deemed suitable for publication in PLOS One. Congratulations! Your manuscript is now being handed over to our production team.

Kind regards,

on behalf of

Dr. Ebenezer Wiafe

Academic Editor

PLOS One